# Topological links in predicted protein complex structures reveal limitations of AlphaFold

Yingnan Hou [1,2], Tengyu Xie[1,2], Liuqing He [1,3], Liang Tao [1,3] & Jing Huang [1,2 ✉]

AlphaFold is making great progress in protein structure prediction, not only for single-chain proteins but also for multi-chain protein complexes. When using AlphaFold-Multimer to predict protein–protein complexes, we observed some unusual structures in which chains are looped around each other to form topologically intertwining links at the interface. Based on physical principles, such topological links should generally not exist in native protein complex structures unless covalent modifications of residues are involved. Although it is well known and has been well studied that protein structures may have topologically complex shapes such as knots and links, existing methods are hampered by the chain closure problem and show poor performance in identifying topologically linked structures in protein–protein complexes. Therefore, we address the chain closure problem by using sliding windows from a local perspective and propose an algorithm to measure the topological–geometric features that can be used to identify topologically linked structures. An application of the method to AlphaFold-Multimer-predicted protein complex structures finds that approximately 1.72% of the predicted structures contain topological links. The method presented in this work will facilitate the computational study of protein–protein interactions and help further improve the structural prediction of multi-chain protein complexes.

[1] Key Laboratory of Structural Biology of Zhejiang Province, School of Life Sciences, Westlake University, 18 Shilongshan Road, Hangzhou 310024 Zhejiang, China. [2] Westlake AI Therapeutics Lab, Westlake Laboratory of Life Sciences and Biomedicine, 18 Shilongshan Road, Hangzhou 310024 Zhejiang, China. [3] Center for Infectious Disease Research, Westlake Laboratory of Life Sciences and Biomedicine, 18 Shilongshan Road, Hangzhou 310024 Zhejiang, China. ✉email: huangjing@westlake.edu.cn

Predicting the three-dimensional (3D) structure of a protein from its primary sequence has long been a topic of great interest in biology. Recent studies using end-to-end deep neural network (DNN) methods such as AlphaFold[1,2] and RoseTTAFold[3] have made great progress in this field. Reliable structure prediction of single-chain proteins has further inspired and facilitated the prediction of the structural details of protein–protein interactions (PPIs)[3,4], which is pivotal for both the understanding of biological functions and intervention in diseases. Several recent studies have presented methods to predict the component identities and interaction modes of PPIs that are built upon AlphaFold[5–8].

Recently, we used AlphaFold-Multimer (v2.2.0)[4] to study PPIs and generated large-scale datasets of predicted protein–protein complex structures. We found that some of the predicted complex structures, including the top-ranked ones with the highest confidence scores, contained unusual topologically intertwining links formed within short backbone fragments in chains at the interface (some examples are shown in Fig. 1a–d). Under physiological conditions, folded proteins encounter each other physically to form interacting complexes with conformational changes but do not unfold. However, the formation of this kind of topological links in protein complex structures requires the unfolding of protein chains, which is nearly impossible to appear in the experimental structures. In nature, topological links in protein complexes can be observed, but they always involve covalently modified amino acids or disulfide bonds, such as in the structures of virus capsids[9,10], or in the intrinsically disordered proteins[11]. We thus believe that the special topological links we observed in the predicted complex structures are likely artifacts generated from the prediction algorithm.

Topologically complex elements such as knots, slipknots, lassos and links have been well studied for single-chain protein structures[12–16] including predicted ones[17–19], which are considered to be associated with particular thermodynamic and kinetic properties[20]. Algorithms that identify and classify the topological features of these nontrivial protein structures are well established[21–28], and some researchers have attempted to apply them directly to multi-chain protein complexes. The identification of topological features for a set of closed curves is mathematically complete; however, protein chains are open curves with N- and C-termini such that determining how to close the curves represents a major difficulty. Existing strategies for closing a protein chain can be mainly classified into two categories: finding loops (e.g., finding covalent bonds and closing accordingly)[9,10,23,29,30] and creating loops (e.g., closing by connecting the N- and C-termini)[26]. The topological feature of a curve is deterministic, but it may vary according to different ways the curve is closed. Therefore, the key step is to create properly closed loops in a protein chain without changing its original topological feature. To this end, approaches such as the KMT algorithm[14,31], minimal surface analysis[32] and the Gauss linking integrals (GLN) method[33] are commonly used to determine the topological types and locations for protein structures with "closed" loops.

For topological link detection in multi-chain protein complex structures, determining how to close at least two curves without changing the original topological features is much more challenging. The LinkProt database[34] established a systematic classification of the topological links in protein structures with three categories: deterministic links, probabilistic links and macromolecular links. Deterministic links and macromolecular links are both topological links in structures that already have loops closed by covalent bonds (e.g., disulfide bonds); for such structures, there is no need to handle the chain closure problem. Probabilistic links are topological features in probabilistic form for structures with random closures, circumventing the chain closure problem. Another method of measuring the entanglement of protein complexes was established by observing the behavior of chains when pulling at both termini of each chain[35], where the closure of the chains and the results of identification are determined by the pulling directions. In addition, the Gauss discrete integral over open chains was suggested for measuring the entanglement in domain-swapped protein dimers without the need for closing curves[36], which is essentially an approximate estimation of the topological feature of two closed chains formed by directly connecting the N- and C-termini of each chain.

As the number of available experimental protein–protein complex structures is limited, existing methods have only focused on the global topological feature for the whole structure when detecting entanglement in multichain protein complexes. Currently, the explosively growing number of predicted structures of protein–protein complexes, with real or fictitious interactions together with their more complicated interfaces, bring new opportunities and challenges for the detection of topological features of protein complexes. We find that the existing methods may fail to identify many topological links observed in the AlphaFold-Multimer-predicted structures. Therefore, there is an

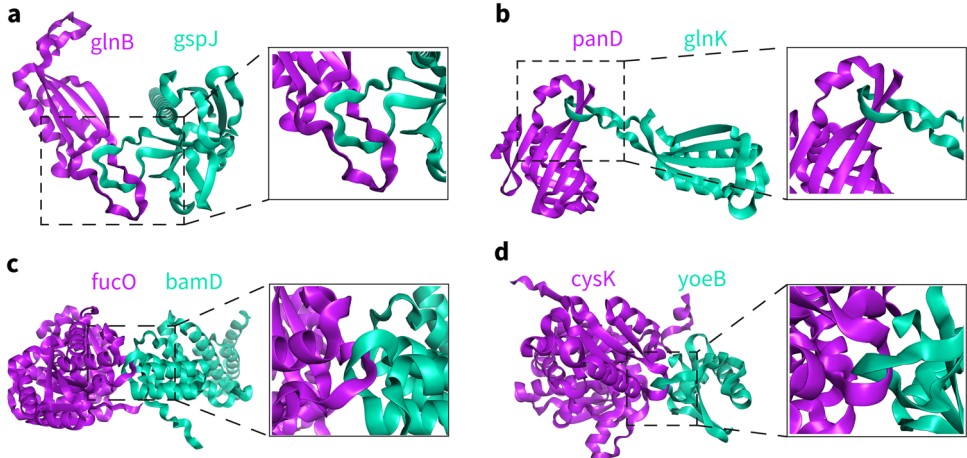

**Fig. 1 Topologically linked structures of protein–protein complexes in *E. coli* predicted by AlphaFold-Multimer (v2.2.0).** The gene names of the proteins are marked. **a**) glnB-gspJ-4 (the fourth predicted structure ranked according to the confidence of AlphaFold-Multimer, same below), **b**) panD-glnK-10, **c**) fucO-bamD-1, **d**) cysK-yoeB-1.

urgent need for methods of identifying topological links in protein-protein complex structures in an accurate and robust manner.

To fill this gap, we propose an efficient method of detecting topologically linked structures in protein complexes, by introducing geometric constraints when measuring topological features from a local perspective. Our method provides a solution to the chain closure problem based on the Gauss integral, by systematically sliding windows in chains instead of focusing on the closures of the termini of chains as in other studies. This is a fast and reliable method with deterministic results regarding how many topological links exist and where the links occur for a given protein-protein complex structure. We demonstrate how this method can detect topologically linked structures in several sets of predicted protein complex structures. Our work may further facilitate the improvement of protein structure predictions and computational PPI studies.

## Results

### Identification of topological links in protein complexes.
To demonstrate why new algorithms are needed to identify topological links in protein-protein complex structures, we applied existing methods to 4 structures predicted by AlphaFold-Multimer v2.2.0 (Fig. 1a–d) and 4 experimental complex structures from the Protein Data Bank (PDB, Fig. 2a–d). For these 4 experimental structures, although two chains are entangled, it is evident that there are no topological links (Fig. 2a–d). However, all 4 experimental structures are indiscriminately identified as topologically linked structures with LinkProt[34] and the GLN method[23], as well as the pulling method[35]. These false inferences are caused by the closure of termini, which creates artificial topological links, indicating the low specificity of the link identification of these methods. For the predicted structures with topological links, LinkProt correctly identified three with Hopf links, but one (Fig. 1b, the 10th predicted structure of *E. coli*

protein panD and glnK complex, named panD-glnK-10 hereafter) was unlinked (Supplementary Table 1). A similar observation was made according to the close-to-zero whole GLN value (0.117). This false unlink inference was also caused by the closure of termini, which eliminated the original links; this happens particularly often for structures containing topological links formed with even-numbered windings in opposite directions in two chains, resulting in a cancellation effect when measuring the topological features from a global perspective. This is also demonstrated by the well-known failure in identifying the Whitehead link by existing methods[23]. The application of those methods on the eight examples illustrates that existing state-of-the-art methods are hampered by the chain closure problem, creating artificial links or eliminating original links, and show low sensitivity and specificity in identifying topologically linked structures in protein-protein complexes.

To overcome the chain closure problem and to detect topological links in protein-protein complex structures, we propose a novel algorithm by introducing an additional geometric dimension when systematically characterizing the topological features from a local perspective. As illustrated in Fig. 3, the algorithm is mainly composed of three steps: interface selection, systematic detection of atoms forming topological links and comprehensive inference. First, the protein chains are simplified by considering only the coordinates together with the covalent bonds of backbone N, Cα, and C atoms, which are represented by an ordered set of atoms. We further isolate the interaction interface by selecting atoms that are within a cutoff distance $D = 10$ Å of each other chain, together with the atoms between them, forming two consecutive open subchains for analysis.

In the second step, we focus on the detection of atoms that form topological links in each chain. We systematically create locally closed loops under geometric constraints, by sliding windows along each chain to create fragments with varying lengths, in which each considered fragment is formed by several

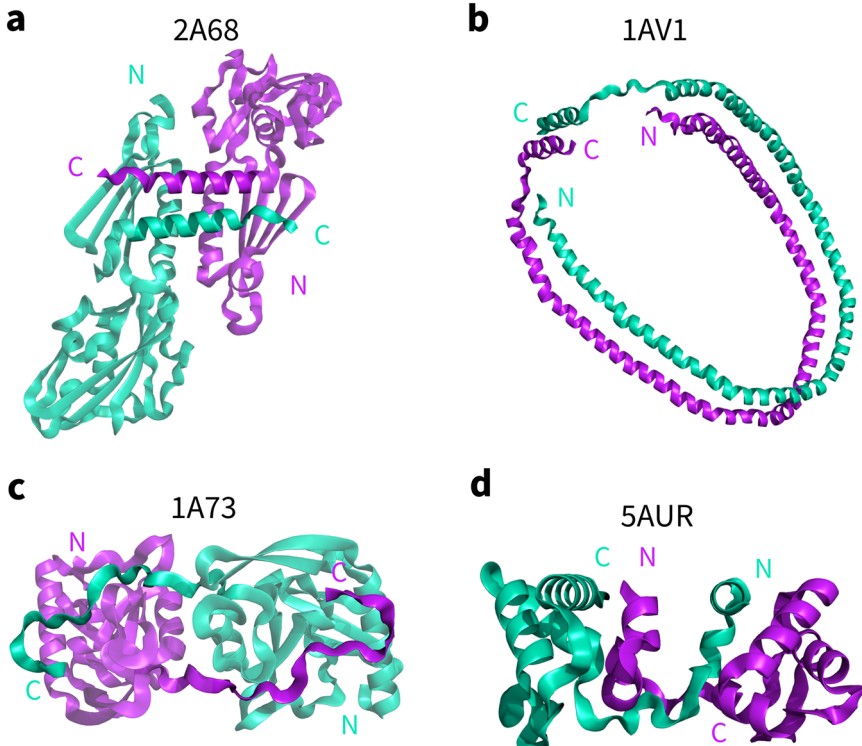

**Fig. 2 Experimental structures of protein–protein complexes in which the chains are wrapped but contain no topological links.** The PDB codes are marked. **a**) 2A68, **b**)1AV1, **c**)1A73 and **d**) 5AUR. N and C represent the N- and C-termini.

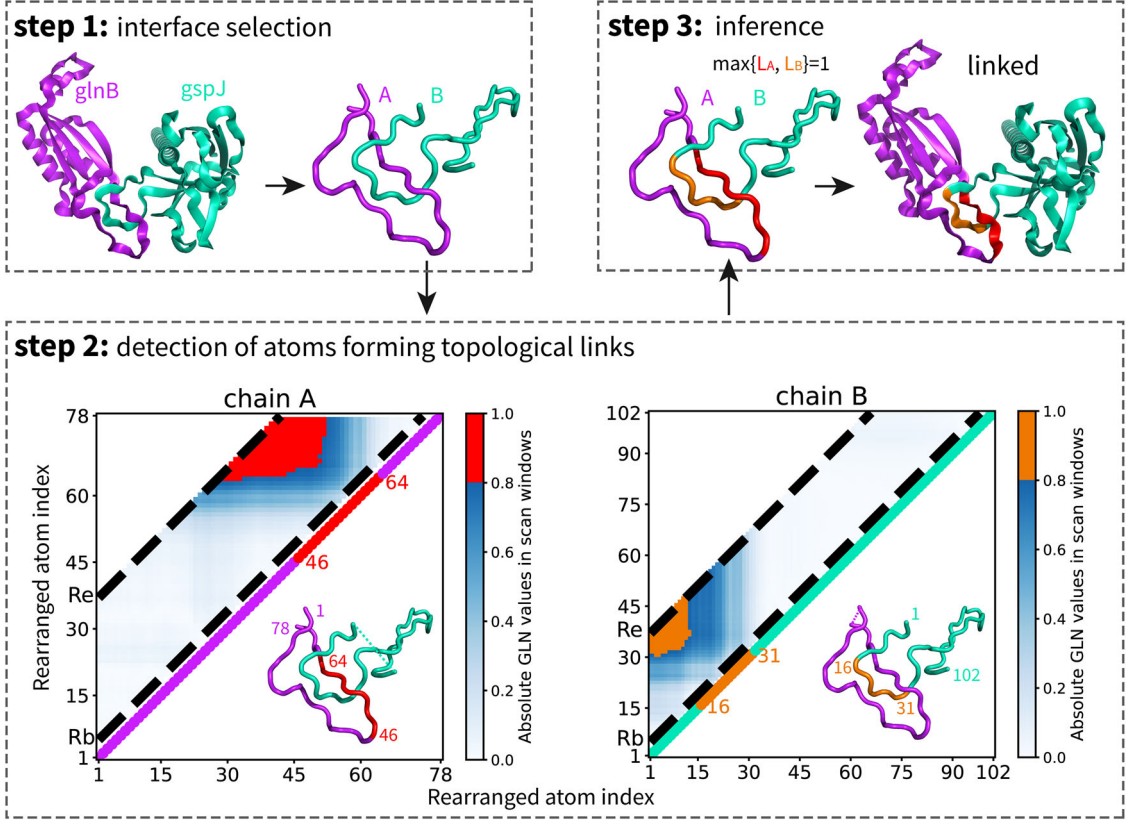

**Fig. 3 Schematic diagram of the algorithm for detecting topological links in protein–protein complex structures.** Step 1: Interface selection to preserve the topological relationships between chains. Step 2: Systematically detection of atoms forming topological links by analyzing the GLN matrixes of both chains. Scanning windows are restricted to fragments whose lengths range between $R_b = 4$ and $R_e = 36$ atoms (the area between the two black dashed lines). Elements with absolute GLN values over $T_s = 0.8$ are highlighted (in red for chain A and orange for chain B), and the corresponding middle atoms are marked to indicate the points at which topological links occur. Step 3: Comprehensive inference of the number of topological links in the structure by using the maximal number of marked fragments in the two chains.

adjacent atoms together with their covalent bonds and is closed by directly connecting its two ends. The topological feature of each fragment in the focused chain with respect to the other chain is measured by the GLN value, and the larger the absolute GLN value is, the more severe the entanglement of the two corresponding curves. All the fragments, each represented by its beginning and ending atoms, correspond to the form of a matrix; thus, the topological features can be measured by the GLN matrix constructed in this way. We restrict the systematic scan to fragments with lengths ranging from $R_b = 4$ to $R_e = 36$ atoms and evaluate the topological features according to the GLN values. Once the corresponding absolute GLN value of a fragment exceeds a threshold score ($T_s = 0.8$), indicating a possible topological link formed between the fragment and the other chain, the middle atom of the fragment, which contributes most to the topological link, will be selected and marked for further analysis in the next step.

In the third step, we infer the number of topological links of the protein complex structure with a comprehensive analysis of the marked atoms. Usually, the same topological link can be identified repeatedly around a set of marked atoms that have covalent bonds with each other, forming a consecutive fragment. Therefore, we calculate the number of consecutive fragments formed by the marked atoms in each chain and use the maximal one as the number of topological links in the structure, which is an indicator of whether a structure is topologically linked. In this way, the algorithm would not only identify whether the complex structure contains topological links but also locate the exact

regions in which they occur. We note that the algorithm is robust with respect to the four user-definable hyperparameters ($D$, $T_s$, $R_e$, and $R_b$).

Analysis of the 8 structures in Figs. 1 and 2 with the present method correctly found topological links for each of the 4 representative predicted structures and no topological links for any of the 4 deeply wrapped experimental structures (Supplementary Table 1), demonstrating its superior ability to identify topological links in protein complex structures. To further illustrate the design logics of those algorithms, we used 3 representative structures, glnB-gspJ-4 (Fig. 1a), panD-glnK-10 (Fig. 1b) and 2A68 (PDB code, chain A and B, Fig. 2b), as examples to demonstrate why the addition of geometric constraints when characterizing topological features (termed topological-geometric features) from a local perspective makes a difference in identifying topological links. As shown in Fig. 4a–i, although glnB-gspJ-4 and 2A68 share the same topological feature (Hopf link) from a global perspective, they show opposite topological–geometric features from a local perspective. Moreover, the cancellation effect of topological links from a global perspective in panD-glnK-10 could be avoided with a local perspective. Those results indicate that topological–geometric features could provide additional key information to characterize protein structures, which can effectively avoid false positives and false negatives in identifying topological links.

We also tested our method with the protein–protein docking benchmark set DB5.0[37], and only one structure (PDB codes: 1NW9) was identified as containing topological links

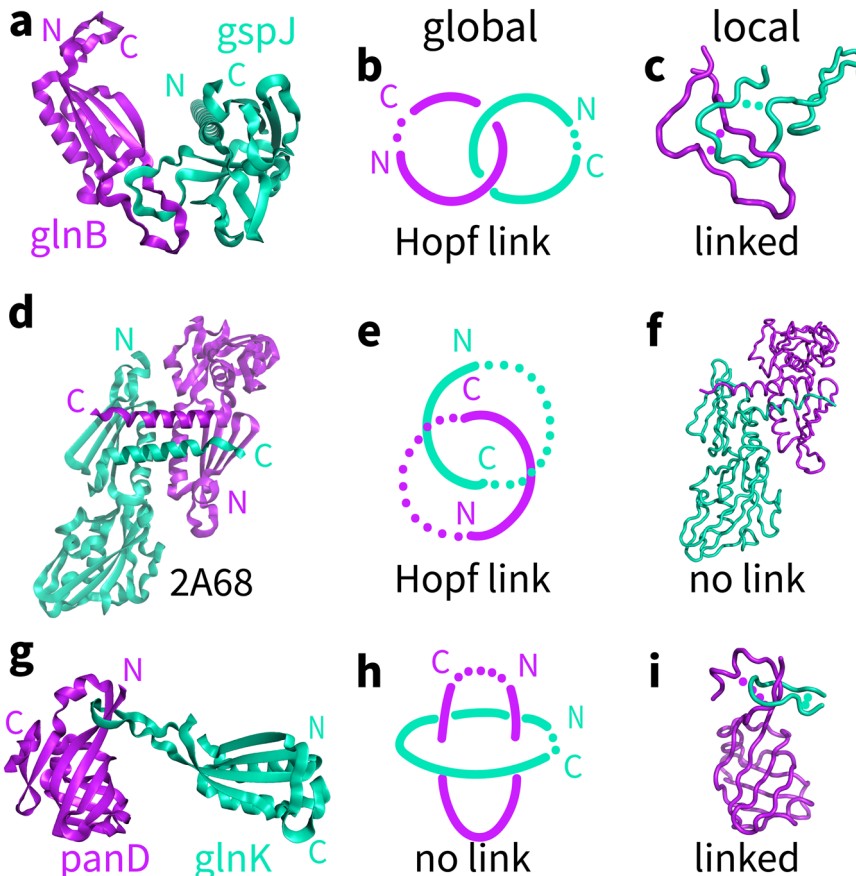

**Fig. 4 Comparison of the differences in characterizing topological features from global and local perspectives. a, d, g)** The representative structures are two predicted structures (**a**: glnB-gspJ-4 and **g**: panD-glnK-10) and one experimental structure (**d**: PDB code 2A68, chain A in purple and chain B in cyan), respectively. **b, e, h)** The topological feature of these protein complexes characterized from a global perspective in which two closed chains are formed by directly connecting the N- and C-termini of each chain. **c, f, i)** The addition of geometric constraints to the topological features in these protein complexes from a local perspective, which create reasonable locally closed loops along each chain.

(Supplementary Data 1). Manual inspection revealed a very low-quality interface structure with an excessive number of missing residues, causing one broken chain at the interaction interface and leading to a false topological link (Supplementary Fig. 1).

We extended our analysis to an additional dataset consisting of 22,003 high-quality experimental protein complex structures from the PDB database after applying two filters requiring 2 chains per assembly and a resolution better than 2 Å. Our method identified 57 structures containing topological links, as detailed in Supplementary Data 2. Upon manual inspection, we found that 19 of these structures have incomplete interaction interfaces with at least 8 missing residues, leading to false topological links. Furthermore, sequence alignment using BLAST confirmed that 17 of these structures contain annotation errors, as the linked two chains are essentially the same chain. We additionally identified 3 structures that contained anomalies in residue or chain indexing. Excluding these structures, we found 18 domain-swapped dimers with deeply entangled chains, some of which coupled with disulfide bonds, resulting in an approximate rate of 0.082%. This analysis further underscores the utility of our algorithm in identifying potential entanglements and links in protein complex structures.

**Applications to protein complex structures predicted with AlphaFold-Multimer.** Our algorithm allows quick identification of topologically entangled structures from a large number of predicted protein–protein complex structures. We used AlphaFold-Multimer to predict protein–protein complex structures for 1,669 experimentally established interacting protein pairs and quantified the number of topologically linked structures. Specifically, we generated two PPI datasets, consisting of 841 protein pairs from Homo sapiens and 828 protein pairs from Drosophila melanogaster, respectively. These datasets were obtained from the positive PPIs benchmark datasets[38] and originally collected from the public Database of Interacting Proteins (DIP) dataset[39], where protein–protein interactions were confirmed through experimental measurements such as two-hybrid screening and co-immunoprecipitation (co-IP)[39]. The sizes of the protein complexes range from 132 to 1,534 residues.

We generated 21,025 and 20,700 predicted structures for the 841 and 828 PPIs using AlphaFold-Multimer (v2.2.0), respectively, with 25 predicted structures for each PPI pair. We applied our method on these datasets and found that 641 and 565 structures contained topological links, respectively (Table 1). Focusing on the structures with the highest model confidence according to AlphaFold-Multimer, i.e., the top-ranked structures among the 25 predictions for each pair, we identified 21 out of the 841 structures (approximately 2.50%) and 16 out of the 828 structures (approximately 1.93%) as containing topologically linked structural elements (see examples in Fig. 5a–b and Supplementary Fig. 2a–b). Besides, the most entangled predicted structure in these datasets contains 10 topological links (Supplementary Fig. 2c).

**Table 1 Summary of link detection by the proposed method on the six sets of AlphaFold-Multimer (v2.2.0) predicted structures.**

| Species | | All predicted structures | | | Top-ranked structures | | |
|---|---|---|---|---|---|---|---|
| Bait proteins | Prey proteins | Total | Linked | Percentage | Total | Linked | Percentage |
| Experimental PPI of *H. sapiens* | | 21025 | 641 | 3.05% | 841 | 21 | 2.50% |
| Experimental PPI of *D. melanogaster* | | 20700 | 565 | 2.73% | 828 | 16 | 1.93% |
| *E. coli* | *E. coli* | 10000 | 60 | 0.60% | 400 | 4 | 1.00% |
| *H. sapiens* | *H. sapiens* | 18000 | 65 | 0.36% | 720 | 3 | 0.42% |
| *H. sapiens* | SARS-CoV-2 | 14350 | 169 | 1.18% | 574 | 5 | 0.87% |
| Measles virus & Human herpesvirus | *H. sapiens* | 6425 | 58 | 0.90% | 257 | 3 | 1.17% |
| Total | | 90500 | 1558 | 1.72% | 3620 | 52 | 1.44% |

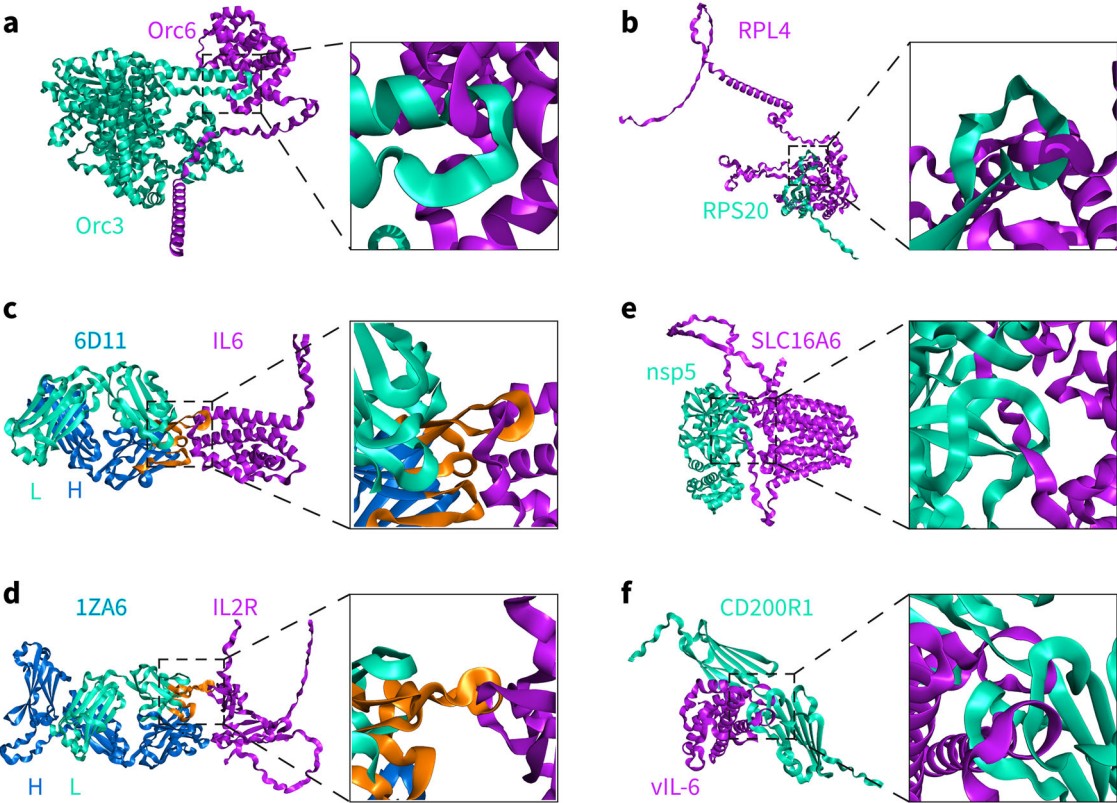

**Fig. 5 Topologically linked structures of protein–protein complexes predicted by AlphaFold-Multimer (v2.2.0). a, b**) Two representative predicted structures (Orc6-Orc3-1 and RPL4-RPS20-1) of PPIs. **c, d**) Two representative predicted structures (6D11-IL6-16 and 1ZA6-IL2R-1) of antibody-interleukin complexes in *Homo sapiens*. The complementarity-determining region (CDR) loops in the chain where topological links occur are colored orange. The PDB codes of the antibodies and the gene names of the interleukins are provided, with L indicating the light chain and H indicating the heavy chain. **e** One representative predicted structure of complexes formed by human MFS transporters and NSPs of SARS-CoV-2 (SLC16A6-nsp5-1). **f** One representative predicted complex structure of pathogenic virus and human membrane proteins (vIL-6-CD200R1-1).

A potential application of AlphaFold-Multimer is to computationally screen candidate PPIs from combinations of proteins with unknown interactions[40,41]. To investigate whether AlphaFold-Multimer predictions for protein complexes with unknown interactions would contain topological linked elements, we generated a dataset by reorganizing protein-protein complexes from some known PPIs in the *Escherichia coli* protein interactome and predicting their structures with AlphaFold-Multimer. We randomly selected 20 PPIs from a benchmark set of the *E. coli* protein interactome[42], with sizes of the complexes between 167 and 1422 residues (Supplementary Data 3). We split the two chains of each of the 20 PPIs into the bait and prey categories following the co-IP terminology, constructed 400 (=20 × 20) paired protein complexes from the two categories and generated 10,000 predicted structures of the 400 protein

complexes using AlphaFold-Multimer. We note that the inference of complex structures was very fast, as the need for MSA was greatly reduced (only 40 proteins involved).

We applied our method to this dataset and found that 60 structures contained topological links (Table 1 and Supplementary Fig. 3a–d). Focusing on the structures with the highest model confidence according to AlphaFold-Multimer, 4 out of the 400 structures were identified as containing topologically linked structural elements (Fig. 1c–d and Supplementary Fig. 3a–b). We confirmed that the 175 predicted structures of the 7 PPIs, whose experimental structures are available at PDB before 2018/5/1 (Supplementary Data 3) and thus may be included in the training set of AlphaFold-Multimer, contain no topological links.

We further tested our algorithm on three other sets of predicted protein-protein complex structures, among which one

set concerns antibody–antigen interactions. Using 40 antibodies with known structures and 18 human interleukins (including IL2 and IL6, details in Supplementary Data 3), we generated 18,000 (40 × 18 × 25) predicted structures of antibody-interleukin complexes with AlphaFold-Multimer. To mimic practical applications in studying host–pathogen interactions, we used the nonstructural proteins (NSPs) of SARS-CoV-2 together with measles virus hemagglutinin glycoprotein (H) and human herpesvirus interleukin-6 homolog protein (vIL-6) to construct protein complexes with human proteins including the major facilitator superfamily (MFS, details in Supplementary Data 3), resulting in two datasets with 14,350 and 6425 structures predicted by AlphaFold-Multimer. We note that in these datasets, most protein pairs will probably not interact under physiological conditions. Applying our method, topologically linked structures were identified in 65 out of 18000, 169 out of 14350, and 58 out of 6425 structures, respectively. For the antibody–interleukin dataset, consisting of flexible human interleukins, and the virus-human datasets, characterized by proteins with very large sizes, the percentages of detected topologically linked structures were consistent with those in the other datasets. For the top-ranked predictions with AlphaFold-Multimer, approximately 0.7% of structures were identified as topologically linked structures in the three sets (Table 1).

For these identified topologically linked structures of predicted antibody–interleukin complexes, we found that all the topological links occur between the antibody chains and the interleukin chain, while none of them occur between the heavy chain and the light chain of the antibody. This is probably due to the larger number of heavy chain–light chain interactions in the training data of AlphaFold. We manually inspected most of the identified topologically linked structures, especially the top-ranked ones, and confirmed the existence of topological links based on the location information provided by the method (Supplementary Fig. 4a–b). In particular, we observed that more than half of the topological links occur around the complementarity-determining region (CDR) loops of the antibodies, as highlighted in orange in Fig. 5c–d and Supplementary Fig. 4b for three representative structures. In addition, there are 2 predicted structures in which the interleukin chains form topological links with both the heavy chain and the light chain of the corresponding antibodies (Supplementary Fig. 5a–b). A recent study also pointed out the limitations of AlphaFold-Multimer in predicting antibody–antigen co-structures[43]. Inspection of the identified topologically linked structures in the other two datasets of virus and human protein complexes show similar results, and some exemplary structures are illustrated in Fig. 5e–f and Supplementary Figs. 6–7.

## Discussion

Proteins physically interact with other proteins to function in many physiological processes, such as signal transduction and immune response. Recent advances in protein structure prediction have opened a new door to computationally determining the structures of protein–protein complexes using only the knowledge of their primary sequences. Currently, predicted protein structures are widely used by biologists. However, the booming number of predicted structures also raises new questions that need to be addressed. Clearly, evaluating predicted structures will be the next research focus after protein structure prediction. In this study, we focus on the topological links observed in predicted complex structures using the current version of AlphaFold-Multimer (v2.2.0) and then focus on detecting them. We developed an algorithm to identify topologically linked structures for protein complexes by providing a solution to the chain closure

problem from a local perspective, that is, systematically sliding windows along each chain instead of closing the N- and C-termini. The algorithm has high sensitivity, as all the topologically linked structures it identified in this work were confirmed by manual inspection. The method runs in seconds per structure, although the exact runtime varies depending on factors such as the size of the protein complex and the degree of entanglement. For example, using a single Intel Xeon E5-2650 v4 CPU core, the average computation time is 8.10 s per complex structure (with an average of 891.2 residues) for the two PPI datasets and 4.37 s per structure (with an average of 721.5 residues) for the *E. coli* dataset, respectively. This makes the method computationally efficient and convenient for large-scale scanning of structures for topological links.

We observed that the confidence scores acquired by AlphaFold-Multimer tend to be relatively low in regions where topological links occur (Supplementary Fig. 8). To investigate the qualitative relationship between the presence of topological links in predicted protein-protein complex structures and the quality of AlphaFold-Multimer predictions, we consider two quality measurements: the overall pTM+iPTM and the average pLDDT score at the interaction interface. As summarized in Supplementary Table 2 and Supplementary Figs. 9–10, no correlation between the number of topological links and the quality of predicted structures was identified. We computed the Pearson correlation coefficients using either quality measurement for each dataset, and all the absolute correlation coefficients were less than 0.06. We confirmed that topological links may occur in structures whose overall pTM+iPTM exceeds 0.8 (Supplementary Fig. 9a). Besides, the average pLDDT scores at the interface range from 17.3 to 90.8 for the predicted topologically linked structures, with 456 structures above 70 (Supplementary Table 2), suggesting that topological links may be present in predicted structures that are believed to be high-quality. Two examples of topological links with confident pLDDT scores at the interface of the predicted structures were shown in Supplementary Fig. 11. We further note that the average interfacial pLDDT can be related to a predicted DockQ score (pDockQ) to distinguish high confidence protein complex structures from incorrect models[6]. When focusing solely on AlphaFold-Multimer predictions with pDockQ > 0.5, 271 out of 5483 and 156 out of 2488 complex structures (4.9% and 6.3%) are identified as containing topological links in the two PPI datasets, respectively (Supplementary Table 3). We also analyzed the distributions of the maximal, minimal, median, and average pLDDT scores of the topologically linked regions within 1,558 AlphaFold-Multimer predicted structures that contains links (Supplementary Fig. 12). Among these structures, we observed that 271 structures (approximately 17.4%) have an average pLDDT value in the topologically linked regions greater than 70.

We further curate a dataset of 306 protein complex structures to benchmark the accuracy of our method and LinkProt; the dataset includes 203 structures with topological links and 103 without. Our method correctly identified 203 linked structures and accurately excluded 100 of the 103 structures lacking topological links, achieving 100% sensitivity and 97.1% specificity (Supplementary Table 4 and Supplementary Data 4). It is important to note that we should not directly compare our method with LinkProt, as LinkProt was not originally designed for this specific task.

In addition to the characterization of the topological features of complex structures, our method provides a measurement for evaluating complex structure prediction programs. For example, AlphaFold-Multimer v2.2.0 generated 1.72% topologically linked structures on average during the prediction of protein–protein complexes, while a previous version (v2.1.0) produced 30.54% topologically linked structures (Supplementary Table 5). This

suggests a notable improvement in structure prediction for multi-chain protein complexes, which can be attributed to the well-documented enhancement in avoiding overly compact models via the added loss function introduced in v2.2.0[4]. Besides, the quality of AlphaFold predictions could potentially be improved using aggressive sampling techniques, such as additional recycles and iterations[44]. However, such an approach was not pursued in this study due to the considerable increase in time and computational resources required for structure generation. Nevertheless, our results demonstrate the systematic persistence of topologically linked structures in AlphaFold-Multimer v2.2.0 predictions, which cannot be eliminated by constraining interfacial residues to prevent crashes, nor through the selection of the most confident (top-ranked) structures from the predictions.

Topological links between subchains of the same protein chain can be formed during the folding process, while topological links between different chains of protein–protein complexes may not represent a physiologically relevant phenomenon. Therefore, we believe that the topologically linked structures of protein complexes predicted by AlphaFold-Multimer occur due to the algorithm's failure to distinguish between inter-chain and intra-chain interactions, indicating an intrinsic flaw in applying current end-to-end structure prediction algorithm, such as AlphaFold, to protein–protein complex structures. This limitation might manifest itself in the prediction of super-large protein complexes and assemblies, and greatly compromise the prediction accuracy. Overcoming this limitation would also be necessary for the further development of computational methods to understand the dynamics and kinetics of protein–protein interactions.

Avoiding topological intertwining in the PPI interface is not trivial for structure prediction[45,46]. The algorithm for capturing the topological–geometric features can be used as an additional loss function to constrain the feasible space in the deep learning network of AlphaFold-Multimer, which may be helpful in reducing the number of abnormal structures during prediction. We hope this research will further facilitate the improvement of protein structure predictions and computational PPI studies.

## Methods

### Algorithm for identifying topological links

*Selection of the interactive interface.* A protein complex with two chains (chain A and chain B) is taken as an example, as a complex composed of $M(\geq 3)$ chains can be split into $M(M-1)/2$ combinations of two-chain complexes. Only backbone heavy atoms (N, Cα and C) together with their covalent bonds are considered, and only atoms within a cutoff distance $D$ of each other's chain together with the atoms between them are preserved; their coordinates are used for further analysis with the indexes rearranged. The first 15 residues from either the N- or C-termini of each chain are removed to reduce the possible impact of their flexibility.

*Detection of atoms that form topological links.* To take a local perspective on topological links and to reduce the computational complexity, one chain (A) is systematically scanned locally in a sliding-window manner, and the other chain (B) is closed by directly connecting its two endpoints. Any fragment formed by adjacent atoms in chain A with a reasonable scanned window length is closed by directly connecting its two endpoints. Together with the simply closed chain B, the detection of the topological feature of the two closed curves is straightforward. The contribution of topological features from the fragment rather than the end-connected part can be represented by the GLN value; the larger its absolute value is, the more severe the winding of the fragment to the closed chain B. Fidelity in link identification is ensured by filtering out inappropriate connections of

fragment ends, which can be achieved by setting a threshold score for the GLN values.

Let $N_1$ and $N_2$ represent the numbers of atoms in chain A and chain B, respectively. Let $S^1 (\in R^{N_1 \times N_1})$ represent the GLN matrix that measures all the fragments of chain A that wind around a closed chain B, where the element $S^1_{k,l} (1 \leq k, l \leq N_1)$ in the $k$th row and the $l$th column of $S^1$, represents the GLN value of the fragment of chain A from the $l$th atom to the $k$th atom winding around chain B. Note that $S^1$ is a lower triangular matrix, i.e., $S^1_{k,l} = 0 (\forall 1 \leq k \leq l \leq N_1)$. Similarly, $S^2 (\in R^{N_2 \times N_2})$, representing the GLN matrix that measures all the fragments of chain B that wind around a closed chain A, will be analyzed in the same way.

To detect the atoms that form topological links, we focus on the fragments of the open chains whose lengths range from $R_b$ to $R_e$ ($0 < R_b < R_e \leq \min\{N_1, N_2\}$). Let $\triangle s^1_{i,j} (R_b \leq j \leq R_e, 1 \leq i \leq N_1 - j)$ represent the absolute GLN values of the fragment from the $i$th atom to the $(i + j)$th atom in chain A, where $\triangle s^1_{i,j} = |S^1_{i+j,i}| = |S^1_{i+j,1} - S^1_{i,1}|$. Here, we state that if $\triangle s^1_{i,j} \geq T_s$, where $T_s$ is an empirical threshold score, then the $\lfloor i+j/2 \rfloor$th atom, i.e., the middle atom of the corresponding fragment, is selected and marked because it may form topological links in chain A with respect to chain B. Note that one atom may be marked several times in fragments of different lengths.

*Calculation of the topological link number.* After a systematic scan of all the fragments in the length interval $[R_b, R_e]$ in each chain, the same topological link can typically be identified repeatedly around a set of marked atoms, which are next to each other and form a consecutive fragment. Therefore, we calculate the number of consecutive fragments formed by the marked atoms in each chain and use the maximal one among all the chains to define the number of topological links in the structure of the protein complex. We call one structure a topologically linked structure if the number of topological links exceeds zero. In addition, the rearranged indexes of the marked atoms are mapped back to the whole structure, and the locations of the topological links in the form of residue indexes are returned by the algorithm.

There are four user-definable hyperparameters in the algorithm, i.e., the upper ($R_e$) and lower ($R_b$) limits of the length range, the threshold score ($T_s$) and the cutoff distance between chains ($D$). The default values ($D = 10$ Å, $T_s = 0.8$, $R_b = 4$ and $R_e = 36$) are empirically set without being elaborately crafted, which seems to be suitable for detecting topologically linked structures on the protein–protein interaction interfaces in this work. We do note that among these four hyperparameters, $R_e$, which is the maximal fragment length in the scanning windows, is the most sensitive, so we smoothed the method by automatically adding 3 to $R_e$ for structures whose maximal absolute GLN value for the searched fragments was less than $T_s$ but greater than $0.9T_s$.

### Dataset generation

The two PPIs datasets were obtained from the positive PPIs benchmark datasets[38] and originally collected from the DIPs[39], which were filtered to include protein pairs with less than 1536 total residues as suggested in[4], and excluded one protein pair whose structural prediction failed. The protein sequences in this work were downloaded from the UniProt database (https://www.uniprot.org/), except that the sequences of the 40 antibodies were taken from PDB (https://www.rcsb.org/), SARS-CoV-2 NSPs were taken from the NCBI database (https://www.ncbi.nlm.nih.gov/), human virus were taken from PDBe (https://www.ebi.ac.uk/pdbe/) and human membrane proteins were taken from the human protein atlas (https://www.proteinatlas.org/). Detailed sequence information is provided in the Supplementary Data 3. The predicted structures of the protein–protein complexes were generated by AlphaFold-

Multimer (v2.2.0). For multiple sequence alignment (MSA) generation, we used UniRef90 v2019_10 (https://ftp.ebi.ac.uk/pub/databases/uniprot/previous_releases/release-2019_10/uniref/), BFD (https://bfd.mmseqs.com/), Uniclust30 v2018_08 (https://wwwuser.gwdg.de/~compbiol/ uniclust/2018_08/), and MGnify clusters v.2018_12 (https://ftp.ebi.ac.uk/pub/databases/metagenomics/peptide_database/2018_12/). The sequence search tools were JackHMMER and HHblits (v3.3.0). For MSA pairing, we used UniProt (https://ftp.ebi.ac.uk/pub/databases/uniprot/current_release/knowledgebase/complete), downloaded on Feb 11, 2022. For the template search, we used "pdb_mmcif" from the PDB database as of Feb 10, 2022. The template search tools were hmmsearch and hmmbuild of HMMER (v3.3.2). All predicted structures were refined using the built-in Amber force field.

**Statistics and Reproducibility**. All quantitative results of topological link detection were presented as integers, with the same outcome when the calculations were repeated at least two additional times. Although the predicted structures generated by AlphaFold-Mulimer for a specific protein sequence may be influenced by random factors, the percentage of predicted structures containing topological links is expected to yield similar results when tests are conducted at the current data scale (in total 90,500 predicted structures).

**Reporting summary**. Further information on research design is available in the Nature Portfolio Reporting Summary linked to this article.

## Data availability
The data used in this work are available at https://github.com/JingHuangLab/topoLink and Supplementary Data 1–4. Supplementary Data 5 is a.xlsx file that includes data for reproducing the GLN matrices in Fig. 3. Additional information is available from the corresponding author upon request.

## Code availability
The source code has been deposited in a GitHub repository (https://github.com/JingHuangLab/topoLink).

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

## Acknowledgements

This work is supported by the National Natural Science Foundation of China (Grant No. 32171247, 21803057 to J.H., and 31970129 to L.T.), the "Pioneer" and "Leading Goose" R&D Program of Zhejiang (2023C03109), the Zhejiang Provincial Natural Science Foundation of China (Grant No. LQ23C050002 to Y.H, LR19B030001 to J.H., and LR20C010001 to L.T.), the Westlake Education Foundation, and Westlake Center for Genome Editing (Grant No. 20200000A992210/001 to L.T.). We thank the Westlake University Supercomputer Center for computational resources and related assistance.

## Author contributions

Y.H. developed the method, wrote the code, performed the calculations, and wrote the initial manuscript. Y.H., T.X. and L.H. collected the protein sequence data, generated and analyzed predicted structures. L.T. revised the manuscript. J.H. designed the project and revised the manuscript. All authors read and approved the final manuscript.

## Competing interests

The authors declare no competing interests.
