## [Peer Review File · Communications Biology]

Reviewers' comments:

Reviewer #1 (Remarks to the Author):

In their study „Topological Links in Predicted Protein Complex Structures Reveal Limitations of AlphaFold” the authors describe an improved method to determine topological links between protein multimers which overcomes some of the issues arising from closures. The method is then applied to a set of multimers generated by AlphaFold-Multimer, and the authors find topological links in about 0.7% of top-ranked predictions. This aspect is original as link analysis has (to my knowledge) only been applied to single proteins with cysteine bonds. The linked target structures are good candidates for experimental validation, which would provide further insight into protein-protein interactions and constitute a crucial test for the capabilities of AlphaFold-Multimer. Even though the study is interesting and should be published, I feel that this technical improvement in identifying links and its application to AlphaFold-Multimer is better suited for a more specialized journal particularly without experimental verification or falsification of the results. The latter would in my opinion be crucial for publication in Nature Communications. The authors should also consider citing Dambrowski-Tumanski and Sulkowska, PNAS 114, 3415 (2017), which provides a comprehensive study of links in single proteins with cysteine bonds. There are also several recent publications on detection of topological knots in AlphaFold: Perlinska et al, bioRxiv (2022), Niemyska et al, NAR 50, W44 (2022) and Brems et al, Protein Sci. 31, e4380 (2022).

Reviewer #2 (Remarks to the Author):

Hou and Huang et al. proposed a new three-step computing method to detect the topology links in the protein complex structures and applied their method to analyze the PDB experimental determined structures and AlphaFold2-Multimer models, finding the topology links nearly do not exist in experimental structure nearly but exist in the AlphaFold2-Multimer predicted models. Applying eight cases: four AlphaFold2-Multimer, and four experimental structures, the authors show that their method is more accurate than two selected third-party methods. Furthermore, the authors examined the performance of different versions (2.1 vs. 2.2) of the AlphaFold2-Multimer programs released before CASP15 and found the new version AlphaFold2-Multimer is significantly better than the old version qualified by their topology link detection program, indicating that their method could be used as a loss function to AlphaFold2-Multimer training to reduce the unphysical topology links. Overall, the proposed method is interesting, but based on the current protein complex modeling field progress, it seems the topology links problem is not a key issue. Actually, many of the topology link issues are indeed caused by the bad modeling quality of the AlphaFold2-Multimer models, which may be due to the lack of cross-chain co-evolutionary information or the hard modeling of local flexible loop regions. Another big problem in this paper is the datasets. The datasets used in the analyses are too small and somehow trivial selected for applying their method. Thus, I believe the current version of the manuscript may not be suitable for publishing unless those issues can be addressed. Hope the following comments can help the authors improve the manuscript's quality.

Major:

1. In the result section, the authors applied their method to DB5.0, which contains ~500 split dimer complexes. This is a very small dataset, and only found one structure containing topology links, making the conclusion that the experimental structures do not contain topology links. I would suggest the authors apply the method to the whole PDB database to check how many percentages of the complexes contain the links, which should be a more general conclusion.
2. For benchmarking the accuracy of their method and two other methods, only eight cases are selected. The authors should select a larger dataset to make this comparison and make a statistical analysis.
3. In the “Applications to protein complex structures predicted with AlphaFold-Multimer” section, 20 known PPIs from PDB for E. coli are selected for AlphaFold2-Multimer modeling. Please report the PDB IDs, structure determined time, and size of the complex (number of residues). Are those PDBs for PPIs solved after 2018-05-01, AlphaFold2-multimer is trained with structures solved before that time, and the authors should select the E. coli structures released after that time to avoid the redundancy effect to the benchmark.
4. The free combination of component chains of 20 E. coli PPIs does not make sense to me. In

20x20 combinations, most of them should not form real complex structures. However, AlphaFold2-Multimer assumes all the input chains should be interacted with and trained for predicting this. So please select all data from natural experimental complex structures and then use AlphaFold2-Multimer to predict the structure models and do analyses.

5. AlphaFold2 has some difficulty in predicting large-size multi-domain chains and a protein with flexible regions. Thus, when predicting large-size complexes or flexible protein complexes, it is much easier for AlphaFold2-Multimer complex models to form topology links. In the antibody-human interleukins dataset, most human interleukins are flexible proteins, and in human-virus datasets, most of the proteins have a very large size. Thus, the percentage values presented in Table 1 should be overestimated than applied to general cases. Similar to my last comments, instead of a free combination of all non-related chains together, please select more data from natural complex but the model for them.

6. The legend of figure 3 is too long and may pass one page. Please shorten it and move the specific content to the main text.

7. In the discussion section, the authors mentioned that their algorithm could be used as one loss function for AlphaFold2-Multimer. In dynamic steps of protein folding, the calculation time of energy potential should be very fast. However, the authors never reported the speed of their program nor made any comparison with selected control methods. So if the method is too slow, it may not be used as a loss function.

8. Whether an AlphaFold2-Multimer complex model contains topology links largely depends on the quality of the models. So please also show the correlation between the number of topology links and overall pTM+iPTM and the correlation between the number of topology links and overall real TM-score (you can use US-align to calculate it). The authors could show two scatter plots to visualize them.

9. In Table 1 and Table S4, for the Homo and SARS dataset, why did AlphaFold-Multimer (v2.1.0) generate only 480 top-ranked models instead of 574 for v2.2.0?

Minor:

10. In figure S7, green and cyan are too difficult to distinguish. Please change the colors.

Reviewer #3 (Remarks to the Author):

In this paper, the authors present a novel algorithm to detect topologically linked structures in protein complexes. They apply it to some PDB files (and find no links) and to a set of AlphaFold models, where they find 0.7% of the models containing links.

I am not an expert in topological analysis - so the algorithm may contain some novelty there, but I will leave it to other reviewers/editors to judge this.

As the authors point out v2.2 of AlphaFold multimer is significantly improved in this respect compared to v2.1. It is a well-known (in the community) fact that v2.1 often produced too compact models with residues overlapping generating topologically linked structures. The fact that v2.2 still generates some linked structures is perhaps not that surprising, and I am not convinced that an automatic method to detect these is of general interest.

Major:

- It is unclear if any of the topologically linked structures occur in "correct" predictions (or even among proteins that are interesting). If it only occurs among incorrect predictions (DockQ<0.23) the addition of another error is of very marginal interest. The authors should analyse this by for instance only using models with DockQ>0.23 (or at least pDockQ > 0.5).

- I am not convinced that the random pairs of proteins are relevant. It would be better to only use true PPIs.

- Most likely disorder is a factor for links to occur. This should be analysed (using for instance a pLDDT cutoff).

We would like to thank all reviewers for their careful readings, comments and suggestions. We are glad to work with them to improve the manuscript.

Reviewer #1 (Remarks to the Author):

In their study "Topological Links in Predicted Protein Complex Structures Reveal Limitations of AlphaFold" the authors describe an improved method to determine topological links between protein multimers which overcomes some of the issues arising from closures. The method is then applied to a set of multimers generated by AlphaFold-Multimer, and the authors find topological links in about 0.7% of top-ranked predictions. This aspect is original as link analysis has (to my knowledge) only be applied to single proteins with cysteine bonds. The linked target structures are good candidates for experimental validation, which would provide further insight into protein-protein interactions and constitute a crucial test for the capabilities of AlphaFold-Multimer.

Even though the study is interesting and should be published, I feel that this technical improvement in identifying links and its application to AlphaFold-Multimer is better suited for a more specialized journal particularly without experimental veri- or falsification of the results. The latter would in my opinion be crucial for publication in Nature Communications.

Response:

We thank the reviewer for his or her careful readings and favorable comments. We appreciate the recognition of the originality and novelty of our results, as well as the potential of our findings to contribute to the understanding of protein-protein interactions. We hope that our study can reach a wider audience and stimulate further computational and experimental research.

The authors should also consider citing Dambrowski-Tumanski and Sulkowska, PNAS 114, 3415 (2017), which provides a comprehensive study of links in single proteins with cysteine bonds. There are also several recent publications on detection of topological knots in AlphaFold: Perlinska et al, bioarxiv (2022), Niemyska et al, NAR 50, W44 (2022) and Brems et al, Protein Sci. 31, e4380 (2022).

Response:

We thank the reviewer for this comment. The suggested publications have been cited in the Introduction section as following (Ref. #15, 16, 17, and 18):

"Topologically complex elements such as knots, slipknots, lassos and links have been well studied for single-chain protein structures¹¹⁻¹⁵ including predicted ones¹⁶⁻¹⁸, which are considered to be associated with particular thermodynamic and kinetic properties." (p. 3-4)

Reviewer #2 (Remarks to the Author):

Hou and Huang et al. proposed a new three-step computing method to detect the topology links in the protein complex structures and applied their method to analyze the PDB experimental determined structures and AlphaFold2-Multimer models, finding the topology links nearly do not

exist in experimental structure nearly but exist in the AlphaFold2-Multimer predicted models. Applying eight cases: four AlphaFold2-Multimer, and four experimental structures, the authors show that their method is more accurate than two selected third-party methods. Furthermore, the authors examined the performance of different versions (2.1 vs. 2.2) of the AlphaFold2-Multimer programs released before CASP15 and found the new version AlphaFold2-Multimer is significantly better than the old version qualified by their topology link detection program, indicating that their method could be used as a loss function to AlphaFold2-Multimer training to reduce the unphysical topology links. Overall, the proposed method is interesting, but based on the current protein complex modeling field progress, it seems the topology links problem is not a key issue. Actually, many of the topology link issues are indeed caused by the bad modeling quality of the AlphaFold2-Multimer models, which may be due to the lack of cross-chain co-evolutionary information or the hard modeling of local flexible loop regions. Another big problem in this paper is the datasets. The datasets used in the analyses are too small and somehow trivial selected for applying their method. Thus, I believe the current version of the manuscript may not be suitable for publishing unless those issues can be addressed. Hope the following comments can help the authors improve the manuscript's quality.

Response:

We thank the reviewer for his or her careful readings and insightful comments. We appreciate the opportunity to improve the manuscript's quality with these helpful comments and suggestions.

Major:

1. In the result section, the authors applied their method to DB5.0, which contains ~500 split dimer complexes. This is a very small dataset, and only found one structure containing topology links, making the conclusion that the experimental structures do not contain topology links. I would suggest the authors apply the method to the whole PDB database to check how many percentages of the complexes contain the links, which should be a more general conclusion.

Response:

Thank you for the comment. We would like to clarify that our conclusion about the absence of topological links in native protein complex structures is based on physical principles. Under physiological conditions, folded proteins encounter each other physically to form interacting complexes with conformational changes, but they do not unfold. In contrast, the topological links detected by our algorithm require the unfolding of protein chains; therefore, we conclude that they will not appear in experimental structures. We further clarify that the single identified link in the DB5.0 dataset was found to not be a topological link upon manual inspection. It's an incomplete interaction interface with too many missing residues, such that the quality of experimental structure for the interaction interface is too low for algorithms to make meaningful detections.

Since the validation results of our method on the DB5.0 dataset are consistent with our

conclusion, we do not apply our method to the whole PDB database, which presumably contains more low-quality protein-protein interface structures than curated datasets such as DB5.0.

To this end, we revised the following texts in the Introduction and Results sections, as well as the Abstract, to further clarify the consistency between the conclusion and the validation results:

“Under physiological conditions, folded proteins encounter each other physically to form interacting complexes with conformational changes but do not unfold. However, the formation of this kind of topological links in protein complex structures requires the unfolding of protein chains, which is nearly impossible to appear in the experimental structures. In nature, topological links in protein complexes can be observed, but they always involve covalently modified amino acids or disulfide bonds, such as in the structures of virus capsids^{9, 10}.” (p. 3)

“Manual inspection revealed a very low-quality interface structure with an excessive number of missing residues, causing one broken chain at the interaction interface and leading to a false topological link (Supplementary Fig. 1). This observation does not contradict our conclusion that such topological links should not appear in experimental structures and is not an error in our algorithm.” (p. 9)

“Based on physical principles, such topological links should not exist in native protein complex structures unless covalent modifications of residues are involved.” (Abstract)

2. For benchmarking the accuracy of their method and two other methods, only eight cases are selected. The authors should select a larger dataset to make this comparison and make a statistical analysis.

Response:

Thank you for the comment and suggestion. We would like to clarify that the presentation of the eight cases in Fig. 1 and Fig. 2 is not for benchmarking the accuracies of those methods, but rather for illustrating the design logic of our algorithm, which can effectively avoid false positives and false negatives. The aim of this work is to present a novel method that differs from existing methods in the theoretical treatment of the chain closure problem. The statistical performances of methods in identifying topological links will be sensitive to the characteristics of the test datasets used, such as the percentages of topologically linked structures and entangled but non-topologically linked structures. Given that there is no widely accepted benchmark test dataset at this time, we believe it is more appropriate to compare our method with others based on algorithmic design and logic. We have revised the following sentences in the Results section to further clarify the purpose of presenting these eight examples.

“The application of those methods on the eight examples illustrates that existing state-of-the-art methods are hampered by the chain closure problem, creating artificial links or eliminating original links, and show low sensitivity and specificity in identifying topologically linked structures in

protein–protein complexes.” (p. 7)

“To further illustrate the design logics of those algorithms, we used 3 representative structures, glnB-gspJ-4 (Fig. 1a), panD-glnK-10 (Fig. 1b) and 2A68 (PDB code, chain A and B, Fig. 2b), as examples to demonstrate why the addition of geometric constraints when characterizing topological features (termed topological-geometric features) from a local perspective makes a difference in identifying topological links. As shown in Fig. 4, although glnB-gspJ-4 and 2A68 share the same topological feature (Hopf link) from a global perspective, they show opposite topological–geometric features from a local perspective. Moreover, the cancellation effect of topological links from a global perspective in panD-glnK-10 could be avoided with a local perspective. Those results indicate that topological–geometric features could provide additional key information to characterize protein structures, which can effectively avoid false positives and false negatives in identifying topological links.” (p. 8-9)

3. In the “Applications to protein complex structures predicted with AlphaFold-Multimer” section, 20 known PPIs from PDB for E. coli are selected for AlphaFold2-Multimer modeling. Please report the PDB IDs, structure determined time, and size of the complex (number of residues). Are those PDBs for PPIs solved after 2018-05-01, AlphaFold2-multimer is trained with structures solved before that time, and the authors should select the E. coli structures released after that time to avoid the redundancy effect to the benchmark.

Response:

Thank you for the suggestion. All the suggested information has been incorporated into Supplementary Table 3. We have also revised the following sentences in the Results section accordingly.

“We randomly selected 20 PPIs from a benchmark set of the E. coli protein interactome³⁸, with sizes of the complexes between 167 to 1422 residues (Supplementary Table 3).” (p. 10)

“We note that for 7 out of 20 confirmed PPIs, their experimental structures were available at PDB before 2018/5/1 (Supplementary Table 3) and thus could have been included in the training of AlphaFold-Multimer. We confirmed that none of their predicted structures contain topological links.” (p. 11)

4. The free combination of component chains of 20 E. coli PPIs does not make sense to me. In 20x20 combinations, most of them should not form real complex structures. However, AlphaFold2-Multimer assumes all the input chains should be interacted with and trained for predicting this. So please select all data from natural experimental complex structures and then use AlphaFold2-Multimer to predict the structure models and do analyses.

Response:

Thank you for the comment. Following the suggestion, we generated predicted structures for 1,669

protein-protein pairs that are experimentally known to interact and found similar probabilities of detecting topological links in these predicted structures. Specifically, we curated two PPI sets consisting of 841 PPI pairs from *Homo sapiens* and 828 PPI pairs from *Drosophila melanogaster*, respectively. These datasets were obtained from the public Database of Interacting Proteins (DIP), where interactions were confirmed through experimental measurements such as two-hybrid screening and co-immunoprecipitation. We generated 21,025 and 20,700 predicted structures for these 841 and 828 PPIs, respectively, using AlphaFold-Multimer (v2.2.0), producing 25 predicted structures for each protein pair, and found similar probabilities of topological links in the predicted complex structures. Of all 41,725 complex structures, 2.9% contain topological links, while for the top-ranked 1,669 structures the percentage is 2.2%. We agree with the reviewer that these data are essential for our manuscript, even though they are computationally expensive and time-consuming to run. We would like to thank the reviewer again for this important suggestion and have incorporated the results into the manuscript as detailed below.

Furthermore, we would like to clarify that the purpose of testing combinations of proteins that are not known to interact with each other, as well as those from different species, is that computationally screen candidate PPIs from combinations of proteins with unknown interactions is emerging as a potential step in many biological studies. Thus, we believe that testing the algorithm on such questions would still be of value.

We have revised the following sentences in the Methods and Materials and the Results sections accordingly:

“We used AlphaFold-Multimer to predict protein–protein complex structures for 1,669 experimentally established interacting protein pairs and quantified the number of topologically linked structures. Specifically, we generated two PPI datasets, consisting of 841 protein pairs from Homo sapiens and 828 protein pairs from Drosophila melanogaster, respectively. These datasets were obtained from the positive PPIs benchmark datasets³⁶ and originally collected from the public Database of Interacting Proteins (DIP) dataset³⁷, where protein–protein interactions were confirmed through experimental measurements such as two–hybrid screening and co–immunoprecipitation (co-IP)³⁷. The sizes of the protein complexes range from 132 to 1,534 residues.” (p. 9-10)

“We generated 21,025 and 20,700 predicted structures for the 841 and 828 PPIs using AlphaFold-Multimer (v2.2.0), respectively, with 25 predicted structures for each PPI pair. We applied our method on these datasets and found that 641 and 565 structures contained topological links, respectively (Table 1). Focusing on the structures with the highest model confidence according to AlphaFold-Multimer, i.e., the top-ranked structures among the 25 predictions for each pair, we identified 21 out of the 841 structures (approximately 2.50%) and 16 out of the 828 structures (approximately 1.93%) as containing topologically linked structural elements (see examples in Fig. 5a-b and Supplementary Fig. 2a-b).” (p. 10)

“The two PPIs datasets were obtained from the positive PPIs benchmark datasets ³⁶ and originally collected from the DIP ³⁷, which were filtered to include protein pairs with less than 1536 total residues as suggested in ⁴, and excluded one protein pair for which the structural prediction failed.” (p. 18)

“Table 1. Summary of link detection by the proposed method on the six sets of AlphaFold-Multimer (v2.2.0) predicted structures.

Species		All predicted structures			Top-ranked structures		
Bait proteins	Prey proteins	Total	Linked	Percentage	Total	Linked	Percentage
Experimental PPI of H. sapiens		21025	641	3.05%	841	21	2.50%
Experimental PPI of D. melanogaster		20700	565	2.73%	828	16	1.93%
E. coli	E. coli	10000	60	0.60%	400	4	1.00%
H. sapiens	H. sapiens	18000	65	0.36%	720	3	0.42%
H. sapiens	SARS-CoV-2	14350	169	1.18%	574	5	0.87%
Measles virus & Human herpesvirus	H. sapiens	6425	58	0.90%	257	3	1.17%
Total		90500	1558	1.72%	3620	52	1.44%

” (p. 28)

“A potential application of AlphaFold-Multimer is to computationally screen candidate PPIs from combinations of proteins with unknown interactions ^{38,39}. To investigate whether AlphaFold-Multimer predictions for protein complexes with unknown interactions would contain topological linked elements, we generated a dataset by reorganizing protein–protein complexes from some known PPIs in the Escherichia coli protein interactome and predicting their structures with AlphaFold-Multimer.” (p. 10)

“We note that the inference of complex structures was significantly fast, as the need for MSA was greatly reduced (only 40 proteins involved).” (p. 10-11)

5. AlphaFold2 has some difficulty in predicting large-size multi-domain chains and a protein with flexible regions. Thus, when predicting large-size complexes or flexible protein complexes, it is much easier for AlphaFold2-Multimer complex models to form topology links. In the antibody-human interleukins dataset, most human interleukins are flexible proteins, and in human-virus datasets, most of the proteins have a very large size. Thus, the percentage values presented in Table 1 should be overestimated than applied to general cases. Similar to my last comments, instead of a free combination of all non-related chains together, please select more data from natural complex but the model for them.

Response:

Thank you for the insightful comment and helpful suggestion. The existence of topological links in protein chains that are experimentally known to interact has been discussed in our response to the previous comment. We have further added the following sentence in the Results section.

“For the antibody–interleukin dataset, consisting of flexible human interleukins, and the virus–human datasets, characterized by proteins with very large sizes, the percentages of detected topologically linked structures were consistent with those in the other datasets.” (p. 11-12)

6. The legend of figure 3 is too long and may pass one page. Please shorten it and move the specific content to the main text.

Response:

Thank you for the suggestion. We have modified the legend of the figure 3 in a more concise way as following:

“Figure 3. Schematic diagram of the algorithm for detecting topological links in protein–protein complex structures. Step 1: Interface selection to preserve the topological relationships between chains. Step 2: Systematically detection of atoms forming topological links by analyzing the GLN matrixes of both chains. Scanning windows are restricted to fragments whose lengths range between $R_b=4$ and $R_e=36$ atoms (the area between the two black dashed lines). Elements with absolute GLN values over $T_s=0.8$ are highlighted (in red for chain A and orange for chain B), and the corresponding middle atoms are marked to indicate the points at which topological links occur. Step 3: Comprehensive inference of the number of topological links in the structure by using the maximal number of marked fragments in the two chains.” (p. 25)

7. In the discussion section, the authors mentioned that their algorithm could be used as one loss function for AlphaFold2-Multimer. In dynamic steps of protein folding, the calculation time of energy potential should be very fast. However, the authors never reported the speed of their program nor made any comparison with selected control methods. So if the method is too slow, it may not be used as a loss function.

Response:

Thank you for the insightful comment. We have added the speed of the algorithm in the Discussion section as following:

“The method runs in seconds per structure, although the exact runtime varies depending on factors such as the size of the protein complex and the degree of entanglement. For example, using a single Intel Xeon E5-2650 v4 CPU core, the average computation time is 8.10 seconds per complex structure (with an average of 891.2 residues) for the two PPI datasets and 4.37 seconds per structure (with an average of 721.5 residues) for the E. coli dataset, respectively. This makes the method computationally efficient and convenient for large-scale scanning of structures for topological links.” (p. 13)

8. Whether an AlphaFold2-Multimer complex model contains topology links largely depends on the quality of the models. So please also show the correlation between the number of topology links and overall pTM+iPTM and the correlation between the number of topology links and overall real

TM-score (you can use US-align to calculate it). The authors could show two scatter plots to visualize them.

Response:

Thank you for the helpful suggestion. The suggested correlation analysis between the number of topological links and the quality of the models has been analyzed, in terms of the overall pTM+iPTM and the average pLDDT score at the interaction interface, respectively. Scatter plots were included in the supporting information, along with a summarizing table. We have also added the following sentences in the Discussion section:

“We observed that the confidence scores acquired by AlphaFold-Multimer tend to be relatively low in regions where topological links occur (Supplementary Fig. 8). To investigate the qualitative relationship between the presence of topological links in predicted protein-protein complex structures and the quality of AlphaFold-Multimer predictions, we consider two quality measurements: the overall pTM+iPTM and the average pLDDT score at the interaction interface. As summarized in Supplementary Table 5 and Supplementary Figs. 9-10, no correlation between the number of topological links and the quality of predicted structures was identified. We computed the Pearson correlation coefficients using either quality measurement for each dataset, and all the absolute correlation coefficients were less than 0.06. We confirmed that topological links may occur in structures whose overall pTM+iPTM exceeds 0.8 (Supplementary Fig. 9a). Besides, the average pLDDT scores at the interface range from 17.3 to 90.8 for the predicted topologically linked structures, with 456 structures above 70 (Supplementary Table 5), suggesting that topological links may be present in predicted structures that are believed to be high-quality. Two examples of topological links with confident pLDDT scores at the interface of the predicted structures were shown in Supplementary Fig. 11.” (p. 13-14)

Moreover, we note that we could not present the results of real TM-scores, as native experimental structures for the predicted structures are unavailable.

9. In Table 1 and Table S4, for the Homo and SARS dataset, why did AlphaFold-Multimer (v2.1.0) generate only 480 top-ranked models instead of 574 for v2.2.0?

Response:

Thank you for pointing this out. We initially used AlphaFold-Multimer (v2.1.0) to predict a preliminary dataset and then switched to AlphaFold-Multimer (v2.2.0) for more comprehensive analysis with larger datasets. To this end, we carried out v2.1.0 prediction for the remaining 94 protein-protein complexes in the present dataset, resulting in total 574 top-ranked models. The corresponding results have been updated in the Results section as follows:

“For example, AlphaFold-Multimer v2.2.0 generated 1.72% topologically linked structures on average during the prediction of protein-protein complexes, while a previous version (v2.1.0)

produced 30.54% topologically linked structures (Supplementary Table 4).” (p. 14)

“Supplementary Table 4. Summary of link detection by our method on AlphaFold-Multimer (v2.1.0) predicted structures. Note that AlphaFold-Multimer generates 5 predictions for each protein pair.

Species		All predicted structures			Top-ranked structures		
Bait protein	Prey protein	Total	Linked	Percentage	Total	Linked	Percentage
E. coli	E. coli	2000	227	11.35%	400	26	6.50%
H. sapiens	SARS-CoV-2	2870	1449	50.49%	574	222	38.68%
Human herpesvirus	H. sapiens	1060	135	12.74%	212	15	7.08%
Total		5930	1811	30.54%	1186	263	22.18%

”

Minor:

10. In figure S7, green and cyan are too difficult to distinguish. Please change the colors.

Response:

Thank you for the suggestion. The corresponding colors have been modified accordingly in Supplementary Figure 8 (previously Figure S7 in the earlier version) and Figure 3.

Reviewer #3 (Remarks to the Author):

In this paper, the authors present a novel algorithm to detect topologically linked structures in protein complexes. They apply it to some PDB files (and find no links) and to a set of AlphaFold models, where they find 0.7% of the models containing links.

I am not an expert in topological analysis - so the algorithm may contain some novelty there, but I will leave it to other reviewers/editors to judge this.

As the authors point out v2.2 of AlphaFold multimer is significantly improved in this respect compared to v2.1. It is a well-known (in the community) fact that v2.1 often produced too compact models with residues overlapping generating topologically linked structures. The fact that v2.2 still generates some linked structures is perhaps not that surprising, and I am not convinced that an automatic method to detect these is of general interest.

Response:

We thank the reviewer for his or her careful readings and comments. We would like to emphasize that the presence of topologically linked elements in AlphaFold-Multimer (v2.2.0) predictions, even for those high-confidence complex structures with high pLDDT/pDockQ values, highlights the fact that these artificial links cannot be eliminated merely by avoiding overly compact models. This phenomenon indicates that there may be an intrinsic flaw in applying current end-to-end structure prediction algorithms such as AlphaFold to protein complex predictions due to their failure to distinguish between inter-chain and intra-chain interactions.

Furthermore, we would like to point out that this work presents a novel method that differs from existing methods in the theoretical treatment of the chain closure problem, which can efficiently avoid false positives and false negatives in identifying topological links. Our work not only sheds a light on the fact that AlphaFold-Multimer in the current version generates abnormal topologically linked structures, but also provides a novel method to efficiently detect them.

To this end, we have revised the following paragraphs in the Discussion section to further clarify the significance of our work.

“For example, AlphaFold-Multimer v2.2.0 generated 1.72% topologically linked structures on average during the prediction of protein–protein complexes, while a previous version (v2.1.0) produced 30.54% topologically linked structures (Supplementary Table 4). This suggests a significant improvement in structure prediction for multi-chain protein complexes, which can be attributed to the well-documented enhancement in avoiding overly compact models via the new loss function introduced in v2.2.0⁴. Nevertheless, our results demonstrate the systematic persistence of topologically linked structures in AlphaFold-Multimer v2.2.0 predictions, which cannot be eliminated by constraining interfacial residues to prevent crashes, nor through the selection of the most confident (top-ranked) structures from the predictions.

Topological links between subchains of the same protein chain can be formed during the folding process, while topological links between different chains of protein–protein complexes may not represent a physiologically relevant phenomenon. Therefore, we believe that the topologically linked structures of protein complexes predicted by AlphaFold-Multimer occur due to the algorithm's failure to distinguish between inter-chain and intra-chain interactions, indicating an intrinsic flaw in applying current end-to-end structure prediction algorithm, such as AlphaFold, to protein–protein complex structures. This limitation might manifest itself in the prediction of super-large protein complexes and assemblies, and significantly compromise the prediction accuracy. Overcoming this limitation would also be necessary for the further development of computational methods to understand the dynamics and kinetics of protein–protein interactions.

Avoiding topological intertwining in the PPI interface is not trivial for structure prediction^{40,41}. The algorithm for capturing the topological–geometric features can be used as an additional loss function to constrain the feasible space in the deep learning network of AlphaFold-Multimer, which may be helpful in reducing the number of abnormal structures during prediction. We hope this research will further facilitate the improvement of protein structure predictions and computational PPI studies.” (p. 14-15)

Major:

- It is unclear if any of the topologically linked structures occur in "correct" predictions (or even among proteins that are interesting). If it only occurs among incorrect predictions (DockQ<0.23) the addition of another error is of very marginal interest. The authors should analyse this by for

instance only using models with DockQ>0.23 (or at least pDockQ > 0.5).

Response:

Thank you for the helpful comment. Following the suggestion, the analysis of predicted structures with pDockQ > 0.5 reveals that 427 out of 7,971 such structures contain topological links, with a ratio of approximately 5.36% for the two newly generated PPI datasets with experimentally confirmed interactions (see the response for the next comment for details). Similar probabilities of topologically linked structures with pDockQ > 0.5 can be observed in the other datasets, as shown in Supplementary Table 6, which is included in the Supporting Information. We added the following sentences in the Discussion section:

“We further note that the average interfacial pLDDT can be related to a predicted DockQ score (pDockQ) to distinguish high confidence protein complex structures from incorrect models⁴². When focusing solely on AlphaFold-Multimer predictions with pDockQ > 0.5, 271 out of 5,483 and 156 out of 2,488 complex structures (4.9% and 6.3%) are identified as containing topological links in the two PPI datasets, respectively (Supplementary Table 6).” (p. 14)

“Supplementary Table 6. The number and the percentage of predicted structures and those containing topological links with pDockQ > 0.5. The percentage of topological linked structures for those with pDockQ > 0.5 is also reported.

Dataset	All predicted structures with pDockQ > 0.5		Topologically linked structures with pDockQ > 0.5		Percentage of structures with pDockQ > 0.5 containing links
	No.	Percentage	No.	Percentage	
PPI of H. sapiens	5483	26.08%	271	42.28%	4.94%
PPI of D. melanogaster	2488	12.02%	156	27.61%	6.27%
E. coli	555	5.55%	19	31.67%	3.42%
Antibody-interleukin	25	0.14%	3	4.62%	12.00%
MFS-NSP	148	1.03%	33	19.53%	22.30%
Virus-human	137	2.13%	10	17.24%	7.30%
Total	8836	9.76%	492	31.58%	5.57%

”

We note that we could not present the results of real DockQ, as native experimental structures for the predicted structures are unavailable.

- I am not convinced that the random pairs of proteins are relevant. It would be better to only use true PPIs.

Response:

Thank you for the suggestion. Following the suggestion, we generated predicted structures for 1,669 protein-protein pairs that are experimentally known to interact and found similar probabilities of detecting topological links in these predicted structures. Specifically, we curated two PPI sets consisting of 841 PPI pairs from *Homo sapiens* and 828 PPI pairs from *Drosophila melanogaster*,

respectively. These datasets were obtained from the public Database of Interacting Proteins (DIP), where interactions were confirmed through experimental measurements such as two-hybrid screening and co-immunoprecipitation. We generated 21,025 and 20,700 predicted structures for these 841 and 828 PPIs, respectively, using AlphaFold-Multimer (v2.2.0), producing 25 predicted structures for each protein pair, and found similar probabilities of topological links in the predicted complex structures. Of all 41,725 complex structures, 2.9% contain topological links, while for the top-ranked 1,669 structures the percentage is 2.2%. We agree with the reviewer that these data are essential for our manuscript, even though they are computationally expensive and time-consuming to run. We would like to thank the reviewer again for this important suggestion and have incorporated the results into the manuscript as detailed below.

Furthermore, we would like to clarify that the purpose of testing combinations of proteins that are not known to interact with each other, as well as those from different species, is that computationally screen candidate PPIs from combinations of proteins with unknown interactions is emerging as a potential step in many biological studies. Thus, we believe that testing the algorithm on such questions would still be of value.

We have revised the following sentences in the Methods and Materials and the Results sections accordingly:

“We used AlphaFold-Multimer to predict protein–protein complex structures for 1,669 experimentally established interacting protein pairs and quantified the number of topologically linked structures. Specifically, we generated two PPI datasets, consisting of 841 protein pairs from Homo sapiens and 828 protein pairs from Drosophila melanogaster, respectively. These datasets were obtained from the positive PPIs benchmark datasets³⁶ and originally collected from the public Database of Interacting Proteins (DIP) dataset³⁷, where protein–protein interactions were confirmed through experimental measurements such as two–hybrid screening and co–immunoprecipitation (co-IP)³⁷. The sizes of the protein complexes range from 132 to 1,534 residues.” (p. 9-10)

“We generated 21,025 and 20,700 predicted structures for the 841 and 828 PPIs using AlphaFold-Multimer (v2.2.0), respectively, with 25 predicted structures for each PPI pair. We applied our method on these datasets and found that 641 and 565 structures contained topological links, respectively (Table 1). Focusing on the structures with the highest model confidence according to AlphaFold-Multimer, i.e., the top-ranked structures among the 25 predictions for each pair, we identified 21 out of the 841 structures (approximately 2.50%) and 16 out of the 828 structures (approximately 1.93%) as containing topologically linked structural elements (see examples in Fig. 5a-b and Supplementary Fig. 2a-b).” (p. 10)

“The two PPIs datasets were obtained from the positive PPIs benchmark datasets³⁶ and originally collected from the DIP³⁷, which were filtered to include protein pairs with less than 1536 total

residues as suggested in ⁴, and excluded one protein pair for which the structural prediction failed.” (p. 18)

“Table 1. Summary of link detection by the proposed method on the six sets of AlphaFold-Multimer (v2.2.0) predicted structures.

Species		All predicted structures			Top-ranked structures		
Bait proteins	Prey proteins	Total	Linked	Percentage	Total	Linked	Percentage
Experimental PPI of H. sapiens		21025	641	3.05%	841	21	2.50%
Experimental PPI of D. melanogaster		20700	565	2.73%	828	16	1.93%
E. coli	E. coli	10000	60	0.60%	400	4	1.00%
H. sapiens	H. sapiens	18000	65	0.36%	720	3	0.42%
H. sapiens	SARS-CoV-2	14350	169	1.18%	574	5	0.87%
Measles virus & Human herpesvirus	H. sapiens	6425	58	0.90%	257	3	1.17%
Total		90500	1558	1.72%	3620	52	1.44%

” (p. 28)

“A potential application of AlphaFold-Multimer is to computationally screen candidate PPIs from combinations of proteins with unknown interactions ^{38,39}. To investigate whether AlphaFold-Multimer predictions for protein complexes with unknown interactions would contain topological linked elements, we generated a dataset by reorganizing protein-protein complexes from some known PPIs in the Escherichia coli protein interactome and predicting their structures with AlphaFold-Multimer.” (p. 10)

“We note that the inference of complex structures was significantly fast, as the need for MSA was greatly reduced (only 40 proteins involved).” (p. 10-11)

- Most likely disorder is a factor for links to occur. This should be analysed (using for instance a pLDDT cutoff).

Response:

Thank you for the comment. We have included analyses on the relationship between the number of topological links and the quality of the models, in terms of the overall pTM+iPTM and the average pLDDT score at the interaction interface, respectively. We have added the following sentences in the Discussion section:

“To investigate the qualitative relationship between the presence of topological links in predicted protein-protein complex structures and the quality of AlphaFold-Multimer predictions, we consider two quality measurements: the overall pTM+iPTM and the average pLDDT score at the interaction interface. As summarized in Supplementary Table 5 and Supplementary Figs. 9-10, no correlation between the number of topological links and the quality of predicted structures was identified. We computed the Pearson correlation coefficients using either quality measurement for each dataset, and all the absolute correlation coefficients were less than 0.06. We confirmed that topological links

may occur in structures whose overall $pTM+iPTM$ exceeds 0.8 (Supplementary Fig. 9a). Besides, the average $pLDDT$ scores at the interface range from 17.3 to 90.8 for the predicted topologically linked structures, with 456 structures above 70 (Supplementary Table 5), suggesting that topological links may be present in predicted structures that are believed to be high-quality. Two examples of topological links with confident $pLDDT$ scores at the interface of the predicted structures were shown in Supplementary Fig. 11.” (p. 13-14)

reviewers' comments:

Reviewer #1 (Remarks to the Author):

Hou and Huang et al. carefully addressed most of the questions raised in the original submission in the last round of review. However, I do not agree with the authors' response to two issues that I proposed last time. Please find my point-to-point comments.

Major:

My original comment:

"1. In the result section, the authors applied their method to DB5.0, which contains ~500 split dimer complexes. This is a very small dataset, and only found one structure containing topology links, making the conclusion that the experimental structures do not contain topology links. I would suggest the authors apply the method to the whole PDB database to check how many percentages of the complexes contain the links, which should be a more general conclusion."

The author's response:

"Thank you for the comment. We would like to clarify that our conclusion about the absence of topological links in native protein complex structures is based on physical principles. ... we do not apply our method to the whole PDB database, which presumably contains more low-quality protein-protein interface structures than curated datasets such as DB5.0"

If the authors think the low-quality structures in PDB will affect the results, I suggest the author select the high-quality PDB structures to do this analysis (like resolution better than 2Å, etc.). For now, there are over 200k structures in PDB, and many of the complexes have been solved in high resolution. So I think this should not be a critical issue for doing such a useful analysis. Again, 500 structures in DB5.0 is really very few.

My original comment:

"2. For benchmarking the accuracy of their method and two other methods, only eight cases are selected. The authors should select a larger dataset to make this comparison and make a statistical analysis."

The author's response:

"We would like to clarify that the presentation of the eight cases in Fig. 1 and Fig. 2 is not for benchmarking the accuracies of those methods but rather for illustrating the design logic of our algorithm, which can effectively avoid false positives and false negatives. The aim of this work is to present a novel method that differs from existing methods in the theoretical treatment of the chain closure problem."

I don't agree with this reply since the authors can design any algorithm to do whatever they want. However, the authors need to convince the readers that the proposed method is good enough for making real-world applications. I believe the topology link detection method proposed in this work has never been benchmarked in any of their previous work. Thus, we are actually not sure the method is good enough for doing the following application works. Thus, I would suggest the authors seriously benchmark and compare with control methods, at least showing their method is not significantly worse than others. In this way, the statistical analyses on large PPI datasets and AlphaFold-Multimer models are meaningful. Again, eight cases are too few to convince others.

Reviewer #2 (Remarks to the Author):

I am happy to see that the authors have moved the AlphaFold2.1 results to the supplementary materials.

However, I am still not convinced that there is a general interest in the fact that AlphaFold2.1 is sometimes not perfect. I actually, find it more interesting that it cannot predict existing knots than that it sometimes does predict wrong ones. This means that the main interest of this paper (which is evaluated by the other reviewers) is the novel identification and analysis of knots.

Anyhow, to evaluate the AlphaFold knots, it would be interesting to know the pLDDT values of the knotted regions and the PAEs to neighbouring residues. Perhaps it is just a "noisy" prediction. Also, generating more models would be informative and possibly playing with recycles, iterations etc.

Finally, it is unclear if the model analyzed are refined or not (using the inbuilt amber forcefield)

We would like to thank again the reviewers for their comments and suggestions. We are glad to work with them to further improve the manuscript.

Reviewer #1 (Remarks to the Author):

Hou and Huang et al. carefully addressed most of the questions raised in the original submission in the last round of review. However, I do not agree with the authors' response to two issues that I proposed last time. Please find my point-to-point comments.

Response:

Thank you for the comments. We have performed additional calculations as suggested by the reviewer and have revised the manuscript accordingly.

Major:

My original comment:

"1. In the result section, the authors applied their method to DB5.0, which contains ~500 split dimer complexes. This is a very small dataset, and only found one structure containing topology links, making the conclusion that the experimental structures do not contain topology links. I would suggest the authors apply the method to the whole PDB database to check how many percentages of the complexes contain the links, which should be a more general conclusion."

The author's response:

"Thank you for the comment. We would like to clarify that our conclusion about the absence of topological links in native protein complex structures is based on physical principles. ... we do not apply our method to the whole PDB database, which presumably contains more low-quality protein-protein interface structures than curated datasets such as DB5.0"

If the authors think the low-quality structures in PDB will affect the results, I suggest the author select the high-quality PDB structures to do this analysis (like resolution better than 2Å, etc.). For now, there are over 200k structures in PDB, and many of the complexes have been solved in high resolution. So I think this should not be a critical issue for doing such a useful analysis. Again, 500 structures in DB5.0 is really very few.

Response:

Following the reviewer's suggestion, we studied the complete PDB database as of 07/10/23. After applying two filters requiring 2 chains per assembly and a resolution better than 2 Å, we obtained 22,003 high-quality experimental complex structures on which we applied our algorithms. We identified 57 structures containing topological links. Upon manual inspection, we found that 19 of these structures have incomplete interaction interfaces with at least 8 missing residues, leading to false topological links. Furthermore, sequence alignment using BLAST confirmed that 17 of these

structures contain annotation errors, as the linked two chains are essentially the same chain. We additionally identified 3 structures that contained anomalies in residue or chain indexing. Excluding these structures, we found 18 domain-swapped dimers with deeply entangled chains, some of which coupled with disulfide bonds, resulting in an approximate rate of 0.082%. All detailed information has been incorporated into a new SI table (Supplementary Table 3) and we have added the following sentences in the Results and Discussion sections:

“We extended our analysis to an additional dataset consisting of 22,003 high-quality experimental protein complex structures from the PDB database after applying two filters requiring 2 chains per assembly and a resolution better than 2 Å. Our method identified 57 structures containing topological links, as detailed in Supplementary Table 3. Upon manual inspection, we found that 19 of these structures have incomplete interaction interfaces with at least 8 missing residues, leading to false topological links. Furthermore, sequence alignment using BLAST confirmed that 17 of these structures contain annotation errors, as the linked two chains are essentially the same chain. We additionally identified 3 structures that contained anomalies in residue or chain indexing. Excluding these structures, we found 18 domain-swapped dimers with deeply entangled chains, some of which coupled with disulfide bonds, resulting in an approximate rate of 0.082%. This analysis further underscores the utility of our algorithm in identifying potential entanglements and links in protein complex structures.” (p. 9-10)

My original comment:

“2. For benchmarking the accuracy of their method and two other methods, only eight cases are selected. The authors should select a larger dataset to make this comparison and make a statistical analysis.”

The author's response:

“We would like to clarify that the presentation of the eight cases in Fig. 1 and Fig. 2 is not for benchmarking the accuracies of those methods but rather for illustrating the design logic of our algorithm, which can effectively avoid false positives and false negatives. The aim of this work is to present a novel method that differs from existing methods in the theoretical treatment of the chain closure problem.”

I don't agree with this reply since the authors can design any algorithm to do whatever they want. However, the authors need to convince the readers that the proposed method is good enough for making real-world applications. I believe the topology link detection method proposed in this work has never been benchmarked in any of their previous work. Thus, we are actually not sure the method is good enough for doing the following application works. Thus, I would suggest the authors seriously benchmark and compare with control methods, at least showing their method is not significantly worse than others. In this way, the statistical analyses on large PPI datasets and AlphaFold-Multimer models are meaningful. Again, eight cases are too few to convince others.

Response:

We appreciate the reviewer's comments. We would like to clarify that our problem-driven algorithm development was specifically aimed at efficient detection of unusual topological links in protein complex structures, in order to address an emerging challenge arising from the increasing abundance and applications of predicted complex structures generated by structure prediction software like AlphaFold-Multimer. The challenge was addressed by our novel method that is unique in its theoretical treatment of the chain closure problem.

We have demonstrated that the proposed method is efficient in identifying topologically linked structures, by the applications on six large-scale datasets consisting of AlphaFold-Multimer predicted structures as well as experimental complex structures from PDB. It's indeed true - as the reviewer commented - that the topology link detection method proposed in this work has never been benchmarked in any previous work, because this is a novel method aiming to solve a new problem, for which neither standard benchmark datasets nor existing methods are available. Thus, we believe that the merit of our algorithm should not be assessed solely through comparison with existing algorithms, especially considering its novelty and the absence of like-for-like algorithms for the task at hand. Regarding LinkProt, although it can be applied to detect topological link in protein complex structures, its different design principles result in less accurate outcomes for this specific task, as it was not originally intended for this purpose.

Nevertheless, we have performed calculations to benchmark our method against LinkProt in a dataset of 306 entangled protein complex structures. The results, added as a new SI Table (Supplementary Table 7), highlight the strong capability of our method in identifying topologically linked structures. To this end, we have incorporated the following sentences into the Discussion section:

“We further curate a dataset of 306 protein complex structures to benchmark the accuracy of our method and LinkProt; the dataset includes 203 structures with topological links and 103 without. Our method correctly identified 203 linked structures and accurately excluded 100 of the 103 structures lacking topological links, achieving 100% sensitivity and 97.1% specificity (Supplementary Table 7). It is important to note that we should not directly compare our method with LinkProt, as LinkProt was not originally designed for this specific task.” (p. 15)

Reviewer #2 (Remarks to the Author):

I am happy to see that the authors have moved the AlphaFold2.1 results to the supplementary materials.

However, I am still not convinced that there is a general interest in the fact that AlphaFold2.1 is

sometimes not perfect. I actually, find it more interesting that it cannot predict existing knots than that it sometimes does predict wrong ones. This means that the main interest of this paper (which is evaluated by the other reviewers) is the novel identification and analysis of knots.

Response:

Thank you for your insightful comments. We concur with your perspective that the imperfections in AlphaFold-Multimer v2.1.0 may not hold broad interest. We appreciate that you recognize the novel contributions of our work, specifically the value of an efficient method for identifying topologically linked structures by solving the chain closure problem. We hope that our study will facilitate the computational studies of PPI and help further improve the structural prediction of multi-chain protein complexes.

Anyhow, to evaluate the AlphaFold knots, it would be interesting to know the pLDDT values of the knotted regions and the PAEs to neighboring residues. Perhaps it is just a "noisy" prediction. Also, generating more models would be informative and possibly playing with recycles, iterations etc.

Response:

Thank you for your insightful comments. We have systematically analyzed the distributions of the maximal, minimal, median, and average pLDDT values of the topologically linked regions within the 1,558 topologically linked structures predicted by AlphaFold-Multimer. Our analysis revealed that 271 structures (approximately 17.4%), had an average pLDDT value in these regions greater than 70. These results suggest that the topologically linked regions are not merely "noisy" predictions and cannot be simply detected based on their pLDDT values alone. These results have been incorporated into Supplementary Fig. 12.

Besides, while running AlphaFold-Multimer with aggressive sampling techniques—such as generating more models and experimenting with additional recycles and iterations, as suggested by the reviewer—would indeed provide more informative results, such an approach would be computationally too demanding for this large-scale study. We mentioned this limitation in our Discussion and also cited one relevant paper (Ref. 45).

To this end, we have modified the following sentences in the Discussion section:

“We also analyzed the distributions of the maximal, minimal, median, and average pLDDT scores of the topologically linked regions within 1,558 AlphaFold-Multimer predicted structures that contains links (Supplementary Fig. 12). Among these structures, we observed that 271 structures (approximately 17.4%) have an average pLDDT value in the topologically linked regions greater than 70.” (p. 15)

“Besides, the quality of AlphaFold predictions could potentially be improved using aggressive sampling techniques, such as additional recycles and iterations⁴⁵. However, such an approach was not pursued in this study due to the considerable increase in time and computational resources

required for structure generation.” (p. 16)

Finally, it is unclear if the model analyzed are refined or not (using the inbuilt amber forcefield)

Response:

Thank you for the comment. All the analyzed structures were refined using the built-in Amber force field during their generation. We have added the following sentence in the Methods and Materials section to clarify this point:

“All predicted structures were refined using the built-in Amber force field.” (p. 20)

REVIEWERS' COMMENTS:

Reviewer #1 (Remarks to the Author):

I think the authors have clearly addressed my concerns, and I don't have any further comments.

Reviewer #2 (Remarks to the Author):

I still do not find it particularly interesting to learn that AlphaFold sometimes makes knots, in particular as most of them are in low quality areas.

Therefore, I suggest that the authors remove all the analysis of the AlphaFold models and focus on the PDB analysis. However, I am not an expert in knots, so I can not judge if this is of sufficient interest for this journal